# High Trunk Truncation as a Potential Sustainable Management Option for Asian Longhorned Beetle on *Salix babylonica*

**DOI:** 10.3390/insects15040278

**Published:** 2024-04-16

**Authors:** Chen Huang, Hualing Wang, Xiaoxia Hai, Zhigang Wang, Fei Lyu

**Affiliations:** 1College of Forestry, Hebei Agricultural University, Baoding 071000, China; huangchenlil@163.com (C.H.); wang_hual@126.com (H.W.); moqi.hai@foxmail.com (X.H.); wzhg432@163.com (Z.W.); 2Hebei Urban Forest Health Technology Innovation Center, Baoding 071000, China

**Keywords:** Asian longhorn beetle, invasive insects, high trunk truncation, eradication

## Abstract

**Simple Summary:**

The Asian longhorned beetle (ALB), *Anoplophora glabripennis* Motschulsky, is a serious wood borer of hardwood trees in North America, Europe, and China, causing substantial economic and ecological losses. Several ALB management strategies have been applied; however, certain drawbacks have been identified with these strategies. To explore effective and environmentally sustainable management options, we investigated the biological characteristics of ALB, including the distribution of frass and emergence holes on trees and preferred branches for ALB foraging and ovipositing. The results showed that 76.31–78.88% of frass holes and 85.08–87.93% of emergence holes were located above a height of 200 cm. Adults showed a preference for feeding on branches 2–3 cm in diameter, and eggs were predominantly laid on 5 cm branches, primarily located above a height of 200 cm. These findings suggest a correlation between the number of ALBs and the height of the tree crown. To test this hypothesis, we investigated whether the number of ALBs decreased when the tree crown was removed above 200 cm (high trunk truncation, HTT). The results revealed a significant decrease in the number after the implementation of HTT. Based on our results we advocate for HTT as an effective ALB management strategy, offering valuable insights into the development of a sustainable approach to controlling the number of ALBs.

**Abstract:**

The Asian longhorned beetle (ALB) causes substantial economic and ecological losses, thus, an environmentally friendly management strategy is needed. Here, we propose high trunk truncation (HTT), the removal of the above 200 cm portion of trees, as a sustainable management strategy to control ALB. To examine the hypothesis, an initial step involved the assessment of various biological characteristics of ALB. Subsequently, a controlled field experiment was carried out utilizing HTT. Finally, HTT was applied in two additional ALB infestation regions. The results of the study of the biological characteristics of ALB showed that 76.31–78.88% of frass holes and 85.08–87.93% of emergence holes were located on branches above 200 cm. Adults preferred to feed on branches 2–3 cm in diameter, ALB eggs were predominantly laid on 5 cm branches, and both were primarily located above 200 cm. These results revealed a correlation between the number of ALBs and the tree crown height. The controlled field experiment showed that the number of ALBs was significantly decreased when the HTT strategy was implemented: approximately 90% of frass holes and 95% of adults were eradicated by HTT compared with the control. Different field surveys involving HTT have shown similar results. These findings provide valuable insights into a sustainable and efficient management strategy for reducing the number of ALBs.

## 1. Introduction

Forests and urban trees provide multiple ecosystem services for city residents, including the provision of food and water resources, as well as timber and nontimber products [1,2]. In these ecosystems, insects, as the most abundant organisms, play a key role in maintaining ecosystem health. However, certain insect species can have detrimental effects on trees, leading to reduced ecological and economic function [1]. This issue is further exacerbated by the widespread distribution of non-native insect species on a global scale [3,4]. The Asian longhorn beetle (ALB), *Anoplophora glabripennis* (Motschulsky), is one of the top 100 most destructive invasive species worldwide and has successfully established itself in various locations across North America and Europe [5,6]. Its capability to kill healthy trees results in significant economic losses for urban parks and rural forests within both native and non-native ALB ranges [5,7]. To date, the strategies employed to combat the ALB have included the eradication of infested trees [7], the application of insecticides to both infested and healthy trees [8,9], and the implementation of biological control through the utilization of natural enemies [10,11,12].

Between 2010 and 2021, more than 45% of eradication plans were successful, but these campaigns were labor-intensive, expensive, and time-consuming [4]. ALB infestations occurring in the Veneto region, specifically in the municipality of Cornuda, northeastern Italy, were initially identified in June 2009 [7]. In response, an extensive and intensive monitoring and eradication program was implemented immediately. From 2009 to 2020, this program involved the establishment of buffer zones covering approximately 7600 hectares, visual inspections of infested host plants or plants within the clear-cut radius, trapping protocols within buffer zones, and public awareness campaigns [7]. In contrast, a substantial 288.4 square miles of infestation area are currently under quarantine for the ALB in various locations, including Worcester County, Long Island, Clermont County, Charleston, and Dorchester counties in the United States [13]. The extensive spread of the infection poses a formidable challenge to achieving successful eradication [7]. In addition, the costs associated with eradication escalate significantly as the area expands, potentially rendering the eradication strategy economically unfeasible [7].

Trunk or soil injections and pesticide sprays with imidacloprid have been applied in China, the United States, and Japan to mitigate the population density of ALB [5,14,15]. In Beijing, China, emamectin benzoate has been injected into the trunks of trees in an infested willow forest as an ALB control measure [8]. Additionally, interventions such as inserting wooden sticks with aluminum phosphide into larval galleries and injecting organophosphate insecticides (methamidophos) into trunks have been utilized to eliminate ALB larvae [9,14]. However, many investigations have highlighted the adverse consequences of pesticide use. One notable example is the reduction in regional insect biodiversity observed in both freshwater and terrestrial systems [16]. Furthermore, the presence of toxic substances within the food web may cause bio-amplification, posing potential risks to human and animal health [16,17,18] and adversely impacting insect pollinators adversely [19,20]. 

Potential biological control agents for ALB include a variety of organisms, including entomopathogenic fungi and bacteria, parasitic nematodes, parasitoids, and predators [21]. In Japan, the entomopathogenic fungi *Beauveria brongniartii* Petch and *Metarhizium brunneum* Petch have already been developed into commercial products, while the latter, *M. brunneum*, is also available for commercial purposes within the United States [21]. These fungi exhibit a high mortality rate against ALB, and the effect of *M. brunneum* fungal infection on accelerated host death is enhanced by neonicotinoid insecticides, as demonstrated in laboratory experiments [22]. Despite their efficacy, the virulence of fungi is constrained by specific environmental temperature conditions [23,24]. To date, fourteen larval parasitoid species have been identified in China and Korea, along with two ectoparasitoids (*Spathius erythrocephalus* Wesmael and *Trigonoderus princeps* Westwood) in Europe and seven braconid parasitoids in North America, that are associated with the ALB in their native and nonnative ranges [6,21,25]. Some larval parasitoids are extreme generalists [10], attacking not only ALB and *A. chinensis* (citrus longhorn beetle, CLB) but also other woodboring pests. However, their mass release as a biocontrol agent in North America has not been considered to date [26], such as might occur with the biocontrol agent *Dastarcus helophoroides* (Fairmaire) [27]. Overall, the abovementioned strategies for the management of ALB have certain drawbacks. Consequently, further investigations into environmentally friendly and economically viable control strategies are critical for effectively managing this pest [6].

This study aimed to develop an environmentally friendly and economically viable control strategy. Understanding the biological characteristics of insect pests is crucial for enhancing insect pest management strategies. Therefore, we examined a variety of biological characteristics, including the distribution of frass and emergence holes on the upper trunks and main branches of *Salix babylonica*, and the ALB foraging and ovipositing preference on branches of different diameters. According to the observed characteristics, we concluded that pruning tree crowns above a height of 200 cm (high trunk truncation treatment) might decrease the number of ALBs. To validate this hypothesis, we conducted a controlled field experiment to analyze the amount of ALB captured across high trunk truncation and no-high trunk truncation treatments in Mancheng, Hebei Province. To validate the efficacy of the control measures, the methodologies were implemented across various regions, and the number of ALBs was assessed by comparing the impacts of high trunk truncation and no-high trunk truncation interventions. The results supported the initial hypothesis, that high trunk truncation treatment could decrease the number of ALBs. We thus propose that high trunk truncation could serve as a viable approach for sustainably managing the ALB on *S*. *babylonica*.

## 2. Materials and Methods

### 2.1. Survey Sites

The survey conducted in this study included a total of three sites, each of which had pure stands of *S. babylonica*, including Mancheng and Gaoyang in Baoding City, and Lingshou in Shijiazhuang City, Hebei Province, China. In Mancheng, *S. babylonica* served as a windbreak and was located adjacent to agricultural fields in BeiXinzhuang (114.43 N, 38.43 E). In Gaoyang city (115.71 N, 38.72 E), *S. babylonica* was used as a landscape tree within a park setting, and in Lingshou (114.44 N, 38.29 E), *S. babylonica* was used as a roadway tree along the sides of roads. The diameter at breast height (DBH) of the sample trees was approximately 18 cm, and the height of the trees was about 600–700 cm. 

### 2.2. Insects 

Wild ALB males and females were obtained from *S. babylonica* trees in Mancheng, in Baoding, Hebei Province, China. The adults were maintained in rearing cages (diameter = 24 cm; height = 22 cm) at densities of six insects (three females and three males) per cage. Insects were provided with fresh *S. babylonica* branches, which were replaced every 2 d. The cages were maintained in the laboratory under controlled conditions at 25 ± 1 °C and 60 ± 10% relative humidity. A natural light/dark cycle of approximately 14 h light/10 h dark (GMT + 8) was used. 

### 2.3. Biological Characteristics of ALB 

#### 2.3.1. Spatial Distributions of Frass and Emergence Holes

To understand the biological characteristics of the ALB, the heights of frass and emergence holes were surveyed in Gaoyang (Zhuanxu Park) and Mancheng (Beixinzhuang), which are located in Baoding City, Hebei Province, China. The data collection was conducted during the spring and summer seasons, specifically from 22 May to 4 July. Three adult observers identified together the fresh frass holes and emergence holes from the ground. The heights of the holes were measured using a telemeter rod (measurement range: 5 m; Hebei Zhufeng Instrument Equipment Co., Ltd., Handan, China) in four orientations (east, west, south, and north). 

#### 2.3.2. Bark Consumption by ALB Adults on Branches with Different Diameters

To determine foraging preferences among branches with different diameters, individual female and male beetles were provided with single branch segments of *S. babylonica* after a 12 h starvation period. Bark consumption experiments were performed in an insect cage (length = 20 cm × height = 8 cm × width = 10 cm) under controlled conditions (25 ± 1 °C and 60 ± 10% relative humidity) (Figure 1A). Branches with different diameters (0.5 cm, 1.0 cm, 2.0 cm, 3.0 cm, and 4.0 cm) were cut into 5 cm-long segments immediately before the experiment and placed in the middle of the cages containing adults. After 24 h, sulfuric acid paper was used to trace the outline of the adult bite marks on the cut branches with a pencil. Subsequently, the tracings of the bite scars were placed over scale paper to calculate the area of the bite scars, representing the area of bark consumed [28]. 

#### 2.3.3. Oviposition Selection on Branches with Different Diameters

Oviposition selection experiments were performed in a cylindrical cage (height = 34.8 cm × diameter = 28.5 cm) under controlled conditions (25 ± 1 °C and 60 ± 10% relative humidity). In this setup, four branches with different diameters (1.0 cm, 2.0 cm, 3.0 cm, and 4.0 cm) and (3.0 cm, 4.0 cm, 5.0 cm, and 6.0 cm) were randomly placed in the four corners of the rectangular acrylic box (length = 20 cm × height = 8 cm × width = 12 cm), and then the rectangular acrylic box was placed inside the cylindrical cage (Figure 1B). To secure the branches, foamed plastic was used to affix the branches in the corners of the rectangular acrylic box. Water was placed in the bottom half of the box to provide a water source for the branches, and foamed plastic obstructed the water to prevent the beetles from falling in. A mated pair of ALBs was introduced into the cage, and a 16–20 mesh nylon gauze net was placed over the cage to prevent insect escape. After 10 days, the bark of the branches was removed, and the numbers of eggs on branches with different diameters were recorded. Ten replicates were measured, respectively.

### 2.4. Control Effectiveness of High Trunk Truncation for ALB in a Controlled Field Experiment

To validate the hypothesis that high trunk truncation treatment could decrease the number of ALBs, approximately 80 *S. babylonica* tree crowns were removed on one side of the road in spring in Mancheng (Figure 1C,D). Conversely, the *S. babylonica* trees on the opposite side of the road were left untreated to serve as the control group (Figure 1C,D). Approximately 30 trees were randomly selected to investigate the number of frass holes per tree within the treatment and control groups in May, one year after high trunk truncation treatments. Additionally, the number of adults per tree was examined on three separate occasions, separated by seven-day intervals, between June and July. The detection of ALB adults was carried out by three adult observers on the ground. The beetles located on the branches below 200 cm were manually captured, and those on branches above 200 cm were poked with fishing poles.

### 2.5. Control Effectiveness of High Trunk Truncation for ALB in Different Areas

To provide additional evidence regarding the efficiency of high trunk truncation treatment in controlling ALB, the practice of high trunk truncation treatment was implemented in another two regions affected by ALB infestation. These regions were Lingshou and Gaoyang, Hebei Province. The total captured beetles before and after high trunk truncation treatments were analyzed in 2014–2018 in Lingshou, Hebei Province. In the spring of 2016, the Landscape Management Bureau in Lingshou, Hebei Province conducted a high trunk truncation procedure on *S. babylonica* trees. In 2014–2018, the total number of beetles captured per collection was recorded during the collection of test insects, with six collection times per year for both 2014 and 2015 and five collection times per year for 2016–2018. Twenty adjacent sample trees were compiled into a group, and then four groups were selected randomly and surveyed each year. The distance between the two groups was at least 20 m. Insects were collected from *S. babylonica* trees at 7–15-day intervals in June–August. In Gaoyang, approximately 40 *S. babylonica* trees were randomly selected to investigate the number of adults per tree in the high trunk truncation treatment and no-high trunk truncation treatment (control) on 29 July and 4 August 2020, respectively. 

### 2.6. Statistical Analysis 

All the statistical analyses were performed using SPSS Statistics v. 21.0 (IBM Corp., Armonk, NY, USA) for Windows. To assess the efficacy of high trunk truncation in *S. babylonica* trees as a control measure for the ALB, the total number of captured ALB adults per time at Lingshou in 2014–2018 was assessed with a Generalized linear model (GLM) with a Poisson distribution and a log link function, followed by Bonferroni’s test (α = 0.05). The numbers of frass holes and adults per tree at Mancheng and Gaoyang were sqrt(1 + x)-transformed to meet the assumptions of Gaussian distribution and homoscedasticity, followed by comparisons using independent-sample *t*-tests. Each measured height was regarded as a data point to calculate the distribution percentages of different height regions in the total number of frass and emergence holes. Then, the distribution percentages of the different height regions in each orientation were regarded as a sample for analyzing the spatial distribution difference of frass and emergence holes using one-way analysis of variance (ANOVA), followed by Bonferroni’s test (α = 0.05). The feeding areas in branches with different diameters were log(1 + x)-transformed to meet the assumptions of Gaussian distribution and homoscedasticity and then compared using ANOVA. The numbers of eggs in branches with different diameters were compared with GLM with a Poisson distribution and a log link function, followed by Bonferroni’s test (α = 0.05).

## 3. Results

### 3.1. Biological Characteristics of ALB

#### 3.1.1. Spatial Distributions of Frass and Emergence Holes

To elucidate the spatial distribution of frass and emergence holes on *S. babylonica* trees, we examined 1047 frass holes and 521 emergence holes in no-pruning tree crowns in Gaoyang and Mancheng, Hebei Province, China (Figure 2 and Figure 3). The frass holes of the ALB larvae were mainly distributed in the *S. babylonica* trees above 200 cm (Figure 2A,B), accounting for approximately 76.30% and 78.88% of the total at Gaoyang and Mancheng, respectively (Figure 2C,D). Within this range, approximately 37.62% and 45.64% of the frass holes were distributed between 201 and 300 cm at Gaoyang and Mancheng, respectively. Frass holes were also present between 301 and 400 cm, making up approximately 32.17% and 29.36% at Gaoyang and Mancheng, respectively (Figure 2C; F = 23.469, *df* = 4, *p* < 0.001 and Figure 2D, F = 46.231, *df* = 4, *p* < 0.001). 

ALB emergence holes were predominantly located in *S. babylonica* trees above a height of 200 cm (Figure 3A,B), accounting for approximately 87.93% and 85.08% of the total holes at Gaoyang and Mancheng, respectively (Figure 3C,D). The majority of emergence holes, about 73.84% and 33.08%, were distributed between 200 and 300 cm at Gaoyang and Mancheng, respectively. Moreover, approximately 14.09% and 34.09% of the emergence holes were distributed in the 301–400 cm layer (Figure 3C; F = 140.591, *df* = 3, *p* < 0.001 and Figure 3D, F = 27.526, *df* = 5, *p* < 0.001). 

#### 3.1.2. Bark Consumption by ALB Adults on Branches with Different Diameters 

After being incubated with the branches of *S. babylonica* for a period of 24 h, both the male and female feeding areas displayed significantly larger dimensions, specifically measuring 2.0 cm and 3.0 cm in diameter, in comparison to the diameters of other branches (Figure 4; F_male_ = 6.311, *p* < 0.001; F_female_ = 13.693, *p* < 0.001; all *df* = 4). There were no significant differences in the feeding areas of either males or females among *S. babylonica* branches with diameters of 2.0 cm, 3.0 cm, or 4.0 cm, but the feeding area was significantly greater than that of branches with diameters of 0.5 cm or 1.0 cm.

#### 3.1.3. Oviposition Selection on Branches with Different Diameters

Significant differences were observed in the average number of eggs on branches with different diameters (Figure 5A: X^2^ = 58.000, *df* = 3, *p* < 0.001; Figure 5B: X^2^ = 32.795, *df* = 3, *p* < 0.001). The average numbers of eggs per 10 days were 0, 0.70 ± 0.37, 1.00 ± 0.37, and 4.1 ± 0.35 on branches with diameters of 1 cm, 2 cm, 3 cm, and 4 cm, respectively. The number of eggs laid per female was found to be larger on the branches with a diameter of 4 cm compared to on the branches with other diameters. However, no significant difference was observed in the number of eggs between branches with diameters of 2 cm and 3 cm (Figure 5A). On the branches with 3 cm, 4 cm, 5 cm, and 6 cm diameters, the mean numbers of eggs were 0.50 ± 0.22, 1.30 ± 0.36, 4.10 ± 0.64, and 1.90 ± 0.44, respectively. The number of eggs per female on branches with a 5 cm diameter was also greater than that on branches of other diameters (Figure 5B). 

Based on the aforementioned characteristics, it was deduced that pruning the tree crown beyond 200 cm could lead to a reduction in the number of ALBs. In order to verify this hypothesis, a series of field experiments were conducted.

### 3.2. Control Effectiveness of High Trunk Truncation for ALB in a Controlled Field Experiment

To validate the hypothesis that high trunk truncation treatment could decrease the number of ALBs, a controlled field experiment was conducted to compare the number of frass holes and adults between the high trunk and non-high trunk truncation groups in Mancheng (Figure 1C,D and Figure 6A). The results showed that the average number of frass holes per tree was approximately 0.53 ± 0.12 when the tree crown underwent high trunk truncation, in contrast to 5.70 ± 0.25 per tree in the absence of high trunk truncation; approximately 90% [(5.70 − 0.53)/5.70] of the frass holes were eradicated by high trunk truncation treatment compared with control. Significantly, the number of frass holes per tree differed between the high trunk and non-high trunk truncation groups (Figure 6B; *t* = 20.208, *df* = 58, *p* < 0.001). Additionally, the number of adults per tree exhibited a significant difference between the high trunk and noon-high trunk truncation treatments (Figure 6C; 5.129 < *t* < 7.691, *df* = 62, *p* < 0.001). Specifically, the average number of adults per tree ranged from approximately 0.06 ± 0.06 to 0.09 ± 0.05 when the tree crown underwent high trunk truncation, while it varied from 1.25 ± 0.24 to 1.72 ± 0.23 per tree in the non-high trunk truncation treatment (Figure 6C); approximately 95% [(1.72 − 0.09)/1.72] of the adults specimens were eradicated by high trunk truncation treatment compared with the control.

### 3.3. Control Effectiveness of High Trunk Truncation for ALB in Different Areas

To ascertain the effectiveness of the control measures, the methodologies were implemented in different regions (Lingshou and Gaoyang), and the number of ALBs was evaluated through a comparison of the impacts of high trunk truncation and non-high trunk truncation interventions. In the years 2014–2015, the recorded total number of captured ALB adults in Lingshou, Hebei Province, was 132.83 ± 2.76 and 131.67 ± 7.71 per 80 trees, respectively (Figure 7A). Following the high trunk truncation of *S. babylonica*, a significant reduction in the number of captured beetles occurred, with 28.40 ± 2.46, 37.75 ± 3.78, and 37.00 ± 1.14 beetles captured per 80 trees in 2016, 2017, and 2018, respectively (Figure 7A). Significant differences were observed in the total number of captured beetles across different years, with the total number in 2014 and 2015 exceeding that in 2016–2018 (Figure 7A; X^2^ = 734.055, *df* = 4, *p <* 0.001). In Gaoyang, the average number of adults per tree was approximately 0.37 ± 0.13–0.43 ± 0.13 when the tree crown underwent high trunk truncation, whereas it was 0.93 ± 0.15–1.15 ± 0.17 per tree in the absence of high trunk truncation. The number of adults per tree differed significantly between the high-trunk and non-high-trunk truncation treatments (Figure 7B; 2.714 < *t* < 3.795, *df* = 78, *p* < 0.008).

## 4. Discussion

The ALB is the most serious nonnative invasive species in Europe and North America [5]. Its threat is considerable, as it can target 209 species or cultivars of healthy trees, out of which 101 species exhibit a higher susceptibility, resulting in substantial economic and ecological losses [4,6,15,29,30]. Although eradication programs have achieved a success rate of more than 45%, the task of addressing larger infestation areas continues to present a significant challenge [4]. In our current investigation, a noteworthy revelation emerged: the application of high trunk truncation treatments led to the removal of 90% of the frass holes and 95% of the adults in the initial year, compared to the non-high trunk truncation treatment (Figure 6B,C). This discovery has promising implications, suggesting that the management of the ALB may not rely solely on complete eradication but could also involve the strategic pruning of the tree crowns of trees that tolerate it and will regrow. Such an approach has the potential to reduce the number of ALBs.

The successful eradication of the ALB involves a multifaceted approach, including the early detection and identification of infested plants [7,31]. However, challenges are encountered in terms of the efficacy of these eradication measures, especially in larger infestation areas. The costs associated with eradication efforts escalate as the eradication area expands, rendering the operation economically unfeasible in extensively infested regions, particularly when dealing with insects exhibiting pronounced polyphagia [7,32]. For example, a region exceeding 20,000 ha in Worcester (Massachusetts, United States) was identified as being infested, prompting the monitoring of over 5 million trees and the subsequent removal of approximately 34,000 trees by 2015 [7,33]. Therefore, successful eradication faces a significant challenge when confronting extensive areas of infestation. 

To develop an environmentally friendly and economically viable control strategy, it is essential to understand the biological characteristics of insect pests. Over the past decade, the primary focus in combating wood-boring pests has consistently been on eradication methods, particularly through the removal of their preferred hosts based on their biological characteristics [4,7,34]. For example, the approach of removing preferred hosts has been used for the management of ALB and CLB, which rank among the most serious invasive species that pose significant threats to forests and urban trees in both North America and Europe. Nevertheless, significant differences can be observed in the biological characteristics of these two *Anoplophora* species. Notably, CLB exhibits a different pattern, with oviposition primarily occurring on the lower trunk, root collar region, and exposed roots, and larval development concentrated in the lower trunk and roots [4]. Thus, the complete removal of infected trees is warranted for CLB, given the prevalence of oviposition and larval development on the lower trunk and in the root collar region. In contrast, the ALB typically engages in oviposition and larval development on the upper trunk and main branches [4]. This finding is consistent with our findings, which indicate that the frass and emergence holes, as well as the foraging and oviposition preference of the ALB, are predominantly concentrated in tree trunks above a height of 200 cm (Figure 2, Figure 3, Figure 4 and Figure 5). Therefore, a viable alternative to complete plant removal could be the removal of the upper 200 cm portion of the tree (High trunk truncation) for ALB management. This recommendation is supported by our findings, which indicated that 76.31–78.88% of the frass holes and 85.08–87.93% of the emergence holes were located on the trunks of trees above 200 cm (Figure 2 and Figure 3).

When emamectin benzoate trunk injections were used to control ALB in an infested willow forest in China, an impressive 89% reduction in the ALB larval population was observed during the first spring after application [8]. Similarly, the implementation of high trunk truncation targeting the control ALB demonstrated significant outcomes, as evidenced by the elimination of 90% of the frass holes and 95% of the adults within the initial year following treatment, in contrast to the absence of pruning intervention (Figure 6). The effectiveness of chemical and high trunk truncation is equivalent in terms of the reduction in the number of adults following treatment. Therefore, the high trunk truncation strategy could serve as an effective alternative for ALB, potentially replacing chemical pesticide applications, as suggested by Wang [35]. Furthermore, the previous study showed that ALB is primarily observed within windbreaks adjacent to agricultural landscapes [4,5]. Reducing chemical pesticide use in windbreaks adjacent to agricultural landscapes can significantly increase the abundance of *Propylea japonica* (Thunberg) and *Harmonia axyridis* (Pallas), which are natural enemies and so enhance the biocontrol function [36]. Honeydew, a carbohydrate-rich secretion produced by Hemipteran insects, serves as a significant source of carbohydrates for beneficial insects. However, the presence of insecticides in honeydew leads to a decrease in the number of numerous beneficial insects [17]. Therefore, reducing the use of insecticides to control the ALB may also maintain beneficial insects, increasing the biodiversity of farmland and the biocontrol efficiency of natural enemies. 

Furthermore, the presence of Allee effects presents a substantial and persistent limitation for populations with low-density [37]. The control experiments showed that 90% of the frass holes and 95% of the adults were eliminated by high trunk truncation within the initial year following treatment. As a result, management strategies that prove effective may prioritize reducing the population below the Allee threshold instead of attempting to eliminate every individual [31]. According to Branco et al. [4], certain countries may prioritize the implementation of containment measures if eradication efforts continue to yield success over the next decade. Consequently, the adoption of the high trunk truncation management strategy could potentially emerge as a prominent choice for controlling the ALB in the coming years.

In summary, the high trunk truncation strategy significantly reduced the number of ALBs on the tree *S. babylonica* in three different experiment sites. Compared to alternative management methods, this approach offers several advantages for ALB control. First, the number of ALBs showed a substantial decrease in the case of the high trunk truncation treatment compared to the control. Notably, after one year of high trunk truncation in Mancheng, 90% of the frass holes and 95% of the adults were removed, surpassing the control treatment group. Second, high trunk truncation acts as a preventive measure against strong winds breaking or uprooting ALB-infested trees, mitigating the severe damage caused by fallen trees. Third, the biomass of trees can rapidly recover within approximately 3–5 years, as exemplified in fast-growing trees such as *Salix*, *Populus*, and other cultivated species. Moreover, the trees that have undergone high trunk truncation treatment in China include *Platanus orientalis*, *Poplar*, *Willow*, *Acer negundo*, *Platanus orientalis*, and *Sophora japonica* (Appendix A). The high trunk truncation strategy may be useful for managing ALB on the above-mentioned trees. Finally, the high trunk truncation strategy might minimally impact the habitats of other arthropods while concurrently reducing the use of pesticides. Given these advantages, we recommend the adoption of the high trunk truncation strategy as a practical approach for controlling the ALB population in other tree species or regions.

## Figures and Tables

**Figure 1 insects-15-00278-f001:**
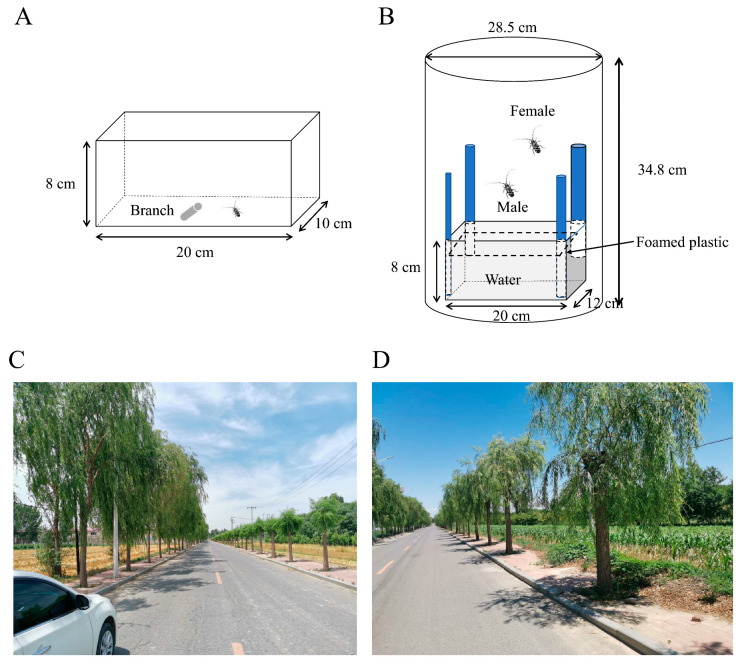
Schematic of the ALB adult feeding (**A**) and oviposition (**B**) behavior bioassay chamber, and the high trunk truncation for *S. babylonica* trees after 1 year (**C**) and 2 years (**D**) in Mancheng, Baoding City, Hebei Province.

**Figure 2 insects-15-00278-f002:**
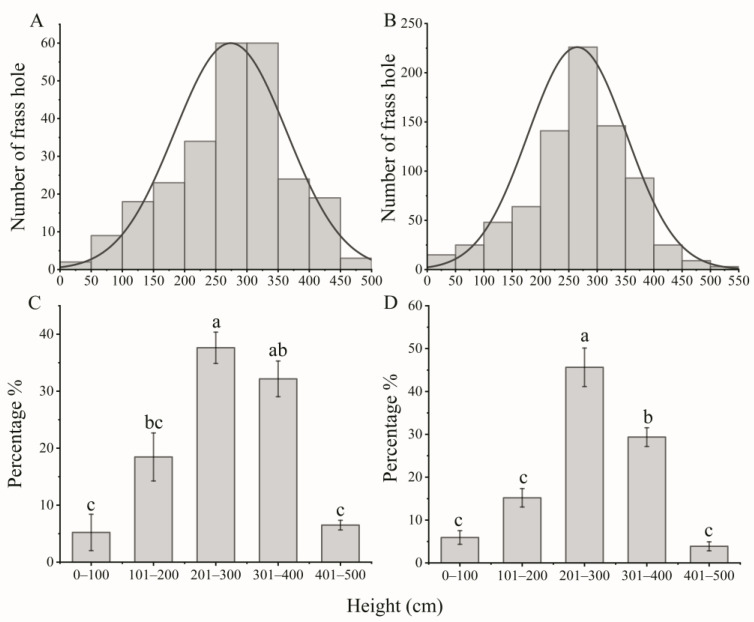
Height distribution conditions of the ALB frass holes in Gaoyang (**A**,**C**) and Mancheng (**B**,**D**), in Hebei Province, China. Count statistics of frass holes at Gaoyang (**A**) and Mancheng (**B**) at different heights. Distribution percentages of frass holes at Gaoyang (**C**) and Mancheng (**D**) at different heights; the percentage indicates the proportion of frass holes at different height ranges of the total number of frass holes. The sample sizes at Gaoyang and Mancheng were 252 and 795, respectively. Differences in the frass holes found in branches of varying diameters were assessed with a one-way analysis of variance, followed by posthoc Bonferroni’s test, α = 0.05. Different letters in the bar indicate that the data are significantly different at α < 0.05.

**Figure 3 insects-15-00278-f003:**
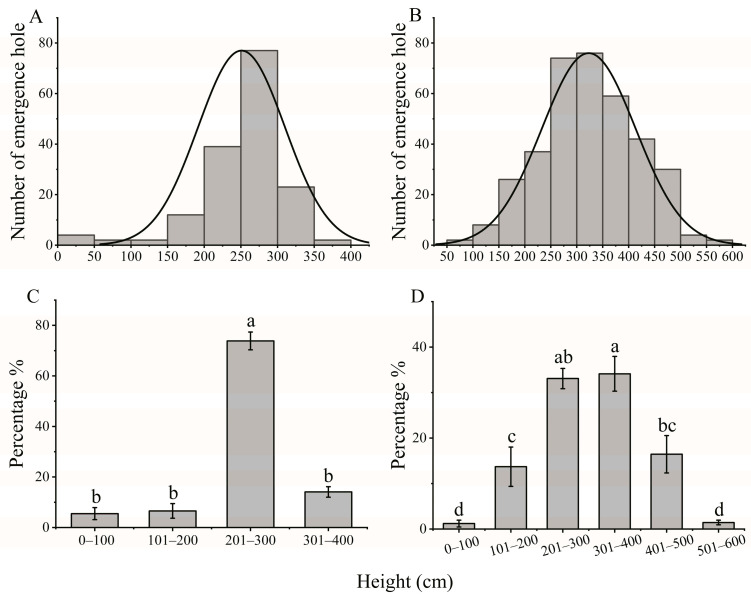
Height distributions of the ALB emergence holes in Gaoyang (**A**,**C**) and Mancheng (**B**,**D**), Hebei Province, China. Count statistics of emergence holes at Gaoyang (**A**) and Mancheng (**B**) at different heights. Distribution percentages of emergence holes at Gaoyang (**C**) and Mancheng (**D**) at different heights. The percentage indicates the relative distribution of emergence holes across various height ranges in relation to the overall number of emergence holes. The sample sizes were 161 and 360 at Gaoyang and Mancheng, respectively. Differences in the emergence holes in branches of different diameters were assessed with a one-way analysis of variance, followed by posthoc Bonferroni’s test, α = 0.05. The different letters in the bar indicate that the data are significantly different α < 0.05.

**Figure 4 insects-15-00278-f004:**
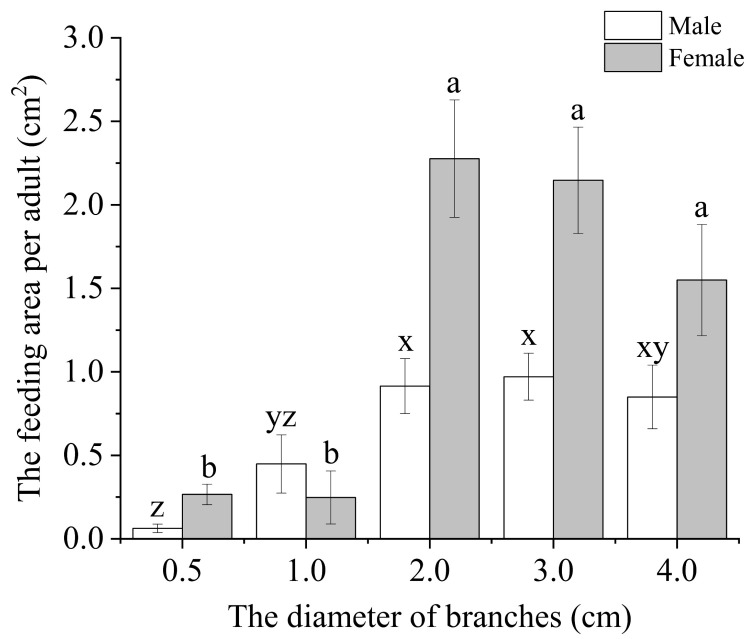
Feeding areas of individual male and female adults per 24 h period on single *S. babylonica* branches. The sample sizes for males and females were 19, 19, 21, 20, and 20 and 19, 19, 18, 21, and 18 for *S. babylonica* branches with diameters of 0.5, 1.0, 2.0, 3.0, and 4.0 cm, respectively. The data are presented as the means ± SEs. Differences in the feeding areas of adults in branches of different diameters were assessed with a one-way analysis of variance, followed by posthoc Bonferroni’s test, α = 0.05. The different letters in the bar indicate that the data are significantly different.

**Figure 5 insects-15-00278-f005:**
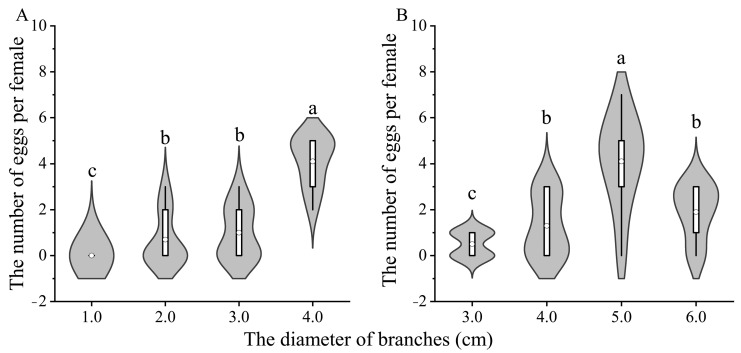
Number of eggs per pair of adults on *S. babylonica* branches in different treatment groups, 1.0, 2.0, 3.0 and 4.0 cm diameter of branches (**A**) and 3.0, 4.0, 5.0 and 6.0 cm diameter of branches (**B**). Differences in the number of eggs laid by a female ALB on branches with different diameters were assessed with a GLM link passion distribution, and pairwise comparisons were conducted with Bonferroni’s test, α = 0.05. Different letters in the violin figure indicate that the data are significantly different. The number of samples was 10.

**Figure 6 insects-15-00278-f006:**
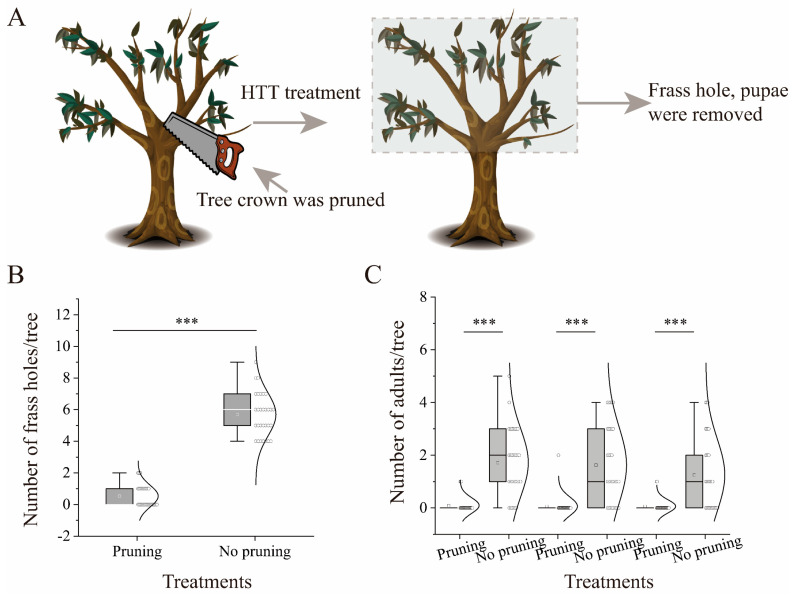
Schematic of high trunk truncation for *S. babylonica* (**A**), number of fresh frass holes (**B**), and number of adults (**C**) per tree in the high trunk truncation and non-high trunk truncation treatments in Mancheng. HTT: high trunk truncation. The upper and lower end points of the vertical lines and the lines of the upper and lower edges of the box indicate the maximum and minimum and 25% and 75%, respectively; the square indicates the mean, and the line indicates the median in the box, the same as below. *** *p* < 0.001 by independent-samples *t*-test. Thirty trees per treatment were investigated in the different treatments in the frass hole experiments, and 32 trees per treatment were investigated in the different treatments in the number of adult experiments.

**Figure 7 insects-15-00278-f007:**
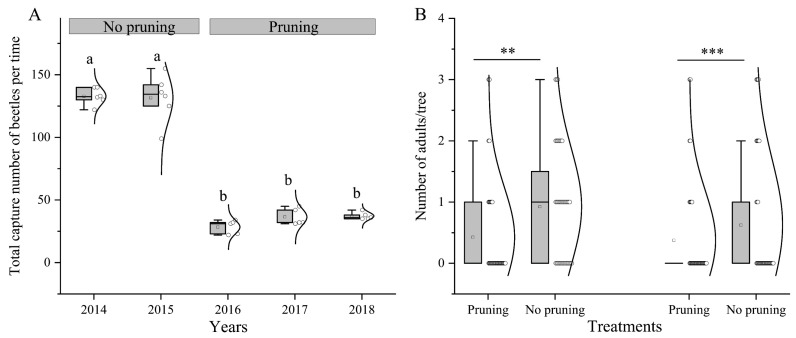
Total number of captured ALBs per time in Lingshou in 2014–2018 (**A**) and the number of adults per tree in Gaoyang in 2020 (**B**). Differences in the number of ALBs were assessed with a GLM linked to a Poisson distribution, followed by Bonferroni’s test, α = 0.05 in Lingshou in 2014–2018. The same letter above the box indicates no significant difference in the number of captured beetles among the different years. Test samples were collected six times per year in both 2014 and 2015, and five times per year in 2016, 2017, and 2018. ** *p* < 0.01, *** *p* < 0.001 by independent-samples *t*-test (n = 40).

## Data Availability

All data supporting the findings of this study are available within the article. Source data are provided with this paper after publication.

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
