# Peer review of "High Trunk Truncation as a Potential Sustainable Management Option for Asian Longhorned Beetle on Salix babylonica"

_insects, 2024, doi:10.3390/insects15040278_

Round 1

Reviewer 1 Report

Comments and Suggestions for Authors

The authors provide useful information on the habits and utilization of Salix babylonica by ALB. However, there are several issues with this paper.  First, there are English grammar and sentence structure issues in at least half of the sentences which need to be fixed. Second, the authors did not provide enough detail in the methods, and this makes interpreting the results difficult. Third, some of the measures used to assess the effectiveness of the treatments are not appropriate. Fourth, the conclusions are not all supported by the results.  Finally, the applicability of this management methods for other hosts and in other settings is very questionable. If the authors do a major revision to correct most of the issues, there will likely still remain a couple that may require dropping small sections of the paper to make it publishable.

Here are more specific comments:

Title suggested revision: High trunk truncation as a potentially sustainable management option for Asian longhorned beetle on Salix babylonica

Simple summary:  should include that this insect is also a problem in China where it is native.

Line 12 replace demonstrate certain with have identified

Lines 12-13 replace with environmentally sustainable management options

Big issue from the start:  How tall were the trees that you worked with?  If they were only 400 cm and you took the top 200 cm off that might work for younger trees that can regrow but if the trees are larger than you would only be leaving trunk and no branches which would not be good for the tree.  Size of the trunk should also be a consideration if the goal is wood production.  If the trees are only hedge rows, then it might not be too bad for the trees if they stay smaller which 9 cm trucks would seem to indicate. However, ALB will use trunks if no good branches are available and have been known to use maple trees that are in hedges for development. Also, you may only be able to cut the trees a couple times before they get too big, and regrowth is not really feasible. The ascetic is also a consideration.  Trees that regrow after such a harsh cut will look very bushy and not pleasing to the eye.

Introduction:

Line 48 reduced not reduce  There are multiple English grammar and sentence structure issues throughout the paper that need to be dealt with before this is assessed again after revisions.  I will not point them all out since that is a waste of my time as a reviewer.

Methods:

2.1  What was the purpose for these trees?  Were they all just hedge rows or wind breaks? Does the public care about what the trees look like?

2.3.1 How were the holes marked to ensure that they were not double counted. Was age of exit holes determined to assess how long the infestation was there and how active it was? Did all three observers independently assess each tree? 

2.3.2 Amount consumed is both the area and the bark thickness.  ALB will consume the bark down through the cambium layer so the thicker the bark the smaller the area to get the same amount of food.  Bark thickness should increase with branch diameter so that has to be accounted for in these measurements.

2.4 Why were the treatments all on one side of the road?  The treatments should have been random to account for any differences that may have existed in the trees between the two sides of the road.  What was the point of poking the beetles with a fishing pole?  If the beetles flew then that would change the density in adjacent trees where they flew to. Why were exit hole in the year after treatment not a characteristic that was evaluated? Exit holes would indicate use and ability to develop.

2.5  Were all the trees treated?  If so, how close are the nearest untreated trees that would serve as a beetle source. The number of beetles present on the trees does not equate with the beetle use of the tree.  If all beetles were collected from the trees each time, then there had to be a source of new beetles nearby. So, what you are really assessing is how likely beetles were to land on the treated trees. You have some confounding factors going on.  Cut trees may be more attractive due to volatile compound coming off them.  Cut trees however may not be as attractive is they have a smaller silhouette. As the trees regrow things may also change. It would have been better to count new frass and exit holes to show more or less use for development by the beetle.

Why is beetle density on the trees more important than beetle use of the trees? What is the end goal of the truncation? Is reducing ALB in an area important for some other reason than just reducing use on these trees?

Results:

Lines 321-322  Beetles were not eradicated they just chose to not be on or use the truncated trees. The beetles that were in the trees when cut were destroyed, but you did not count those.

3.3 Were all the trees treated in the area? Beetles will move around.  Comparing between years is problematic since there may be several things that affect beetle population size year to year.

Discussion:

Although crown pruning seemed hold some promise for this Salix species it would not work for very many trees. Most hardwood trees will not regrow like Salix does and landscape trees have a lot of their value in the shade and beauty they provide which this treatment removes.

Lines 373 to 382 The only way that eradication is possible is to remove all trees with signs, so the beetle does not continue to move around and reinfest trees.  By just removing part of the tree and part of the beetles you extend the eradication effort longer.  Also, removal meas you don’t have to inspect that tree again.

Lines 399-400  This finding is only applicable to 9 cm diameter trees that are less than 800 cm tall of the species you investigated.  A full-grown silver maple tree for example would have a large trunk that measures 2000 cm before any branches occur.  Also just removing the upper 200 cm would not cover the same part of the tree in ever age class and species of tree. You need to be much more specific about the applicability of your tested method.

Lines 402 to 424  For your specific trees and locations as wind breaks this may be a good alternative to chemical treatments. But be careful how generally you suggest it be useful.

Lines 425 to 432  The truncation method would not take the beetles to low enough numbers to have the allee effect come into play. Especially if there are other trees in the vicinity that could harbor ALB.

Line 433 You are using population density too broadly here.  In one study you looked at beetle use of the trees and the other live beetles on the trees.  These are two very different measures that evaluate different points in the lifecycle.

Comments on the Quality of English Language

I only provided a couple of instances of English issues at the very beginning of the paper. There are instances throughout that need to be fixed. Fixing them is not a reviewer's job and would be a waste of my time.

Reviewer 2 Report

Comments and Suggestions for Authors

First, I will provide overall comments on this paper.

As the authors state in the Introduction and elsewhere, the control of the Asian longhorn beetle (ALB), a non-native insect species with a widespread distribution, poses numerous challenges.

In this paper, it has been demonstrated that high trunk truncation treatment serves as an effective method instead of complete felling or root removal. The examination to determine the treatment height provides valuable data on the ecological characteristics of ALB. 

 Their investigation into biological characteristics and the methodical approach are appropriate.

On the other hand, while this method proves effective in resilient Salix species, there is a concern for the risk of mortality in other tree species. 

In relation to the above feedback, I will make several additional comments on specific sections. 

Lines 306–359: Authors indicated the effectiveness of high trunk truncation in reducing ALB density. Additionally, it would be beneficial to ascertain whether the method is more efficient compared to others. Therefore, it is suggested to provide information on the time required for treatment, even if it's just rough estimates (for example, total or per-tree treatment time).

Line 72: Salix babylonica is also a host for other Anoplophora species (i.e., CLB) besides ALB. Have you confirmed whether they coexist or not in the treated trees?

Line 176: I am afraid that ten days seemed to be too long. Haven't the females bitten the branches where oviposition occurred as food and destroyed the eggs? In other words, is there a possibility that the eggs were destroyed, resulting in low egg counts, especially on thinner branches?

Line 179: How were the pruned trees disposed? It would be helpful to know about the method of disposal of trees. This is because if they are simply accumulated elsewhere, there is a concern that they might become a new source of infestation. 

I hope these will serve as useful references for refining the manuscript.

Reviewer 3 Report

Comments and Suggestions for Authors

The research paper is very well-written, the methodology is well described and results are well presented. There are only some minor issues that have to be solved to meet the journal's standards for publication.

At line 22 you use the abbreviation HTT wtithout mentioning earlier, fix it in line 20.

In Line 53 use capability instead of capable (or rephrase the whole line)

In Line 63 use "," between the digits in 7,600 (as you do in Lines379 and 381

In Line 96 References should be numbered to follow the journal's standards

In Line 108 you metion S. babylonica without mentioning its whole genus Salix earlier in the manuscript

At last, at some insects species you mention the nomenclator, e.g., Lines 86 & 87, and in some others you do not e.g., Line 94. Please make it all the same for the whole manuscript

Author Response

The research paper is very well-written, the methodology is well described and results are well presented. There are only some minor issues that have to be solved to meet the journal's standards for publication.

At line 22 you use the abbreviation HTT wtithout mentioning earlier, fix it in line 20.

Response: done

Thank you very much for your comment. We have revised relative sections as suggested. (Line 22, page 1 in the marked-up version)

In Line 53 use capability instead of capable (or rephrase the whole line)

Response: done

Thank you very much for your comment. We have revised relative sections as suggested. (Line 56, page 2 in the marked-up version)

In Line 63 use "," between the digits in 7,600 (as you do in Lines379 and 381

Response: done

Thank you very much for your comment. We have revised relative sections as suggested. (Line 68, page 2; Lines393 and 395, page 12; in the marked-up version)

In Line 96 References should be numbered to follow the journal's standards

Response: done

Thank you very much for your comment. We have revised relative sections as suggested. (Line 102, page 3 in the marked-up version)

In Line 108 you metion S. babylonica without mentioning its whole genus Salix earlier in the manuscript

Response: done

Thank you very much for your comment. We have revised relative sections as suggested. (Line 114, page 3 in the marked-up version)

At last, at some insects species you mention the nomenclator, e.g., Lines 86 & 87, and in some others you do not e.g., Line 94. Please make it all the same for the whole manuscript

Response: done

Thank you very much for your comment. We have revised relative sections as suggested. (Lines 100, 106, page 3 in the marked-up version)

***************************************

We are grateful for these invaluable comments from the Editor and reviewers. Thank you very much!

Round 2

Reviewer 1 Report

Comments and Suggestions for Authors

The authors have made substantial revisions which help but do not solve all the remaining issues. The methods are still lacking some necessary details. The results still need some work to make sure everything is properly described and would be better if they were rearranged to match the methods rather than mixing study results together.  The discussion needs a little work too.  Specifically, the authors need to make clear what other trees this might work on and when they only assessed numbers of adults and not other signs of population presence.  Another round of revisions is needed.  Specific comments are below.

Line 28  I think you mean removal of the crown above 200 cm and not just he upper 200 cm of the tree since you noted that the trees were about 600-700 cm tall.

Line 72-73 need to state these infestations are in the united states.  Are there eradication efforts that have been attempted in China?  How successful have they been?

Lines 125-126  managing ALB on Salix babylonica

Line 153  not on the ground but from the ground

Line 168 not convinced that the bark thickness did not play a role in the amount of bark consumed.  Can the authors provide the thickness of the bark for each diameter of bolt so the reader can at least consider this as a possibility?

Line 199-200  the purpose of poking the beetles above 200 cm is still not provided.  If poking them cause them to move or fly they could have gotten double counted.

Counting beetles on each side of the road (two treatments) only provides information on which trees they preferred to land on and does not measure usage or time on the host.  Why did you not assess new exit holes and frass sites after treatment? Those measures would get at use of the two sets of trees.

Were only S. babalonica truncated in 2.5?  Author’s response indicated the following:

“In China, the trees, which were used by high trunk truncation treatment, include Platanus orientalis (A), Poplar (B), Willow (C), Acer negundo (D), Platanus orientalis (E), Salix babylonica (F, G), Sophora japonica (H).”

If no other trees were truncated in this study then you still need to mention what other trees this technique may be useful for managing ALB.

Line 279  not “in” but “with” the branches

Line 314-15 actually a reduction in the utilization of the trees by ALB, how many individuals are growing in the tree and could emerge.  Whether is also reduces the attractiveness of the trees for adults would be a different question. The number of adults on a tree would be a function of how many emerged from it and stayed and the number of adults that were attracted to the tree and landed on it. 

Figure 6 B seems to indicate that you counted the number of new frass holes in the two treatments the year after the treatment.  Is that the case or is this just the estimates of how many frass holes would be left on the trees after the treatments were implemented?  If you counted the year after treatments then this needs to be added to the methods and would strengthen your conclusions. If not specify what you are reporting here.

Figure 6 C need to label which site each set of bars was from.

Figure 7 B is not properly labeled.  What data is this and from what sites? The other pare of the graph is for the Lingshou study.  This seems to be part of the control verses treatment study.

It would be better to have the results section follow the same pattern as the methods section so the reader can know which study a graph is from.

Line 385 Modify the sentence to say : strategic pruning of tree crowns of trees that tolerate it and will regrow.

Line 419 should be above 200 cm and not taller than 200 cm

Line 426-427 Should read: the effectiveness of chemical and high trunk truncation is equivalent in terms of the reduction in the numbers of adults following treatment.

Good that population density was removed in the discussion but it needs to also be removed elsewhere since you only measured the number of adults in the trees for the full site truncation.

Comments on the Quality of English Language

Most of the issues have been fixed but new text still has some and a couple places the wrong word was substituted.
